# Luminescence Properties and Energy Transfer of Eu^3+^, Bi^3+^ Co-Doped LuVO_4_ Films Modified with Pluronic F-127 Obtained by Sol–Gel

**DOI:** 10.3390/ma16010146

**Published:** 2022-12-23

**Authors:** Brenely González-Penguelly, Grethell Georgina Pérez-Sánchez, Dulce Yolotzin Medina-Velázquez, Paulina Martínez-Falcón, Angel de Jesús Morales-Ramírez

**Affiliations:** 1División de Ciencias Básicas e Ingeniería, Universidad Autónoma Metropolitana-Azcapotzalco, Ciencias Básicas e Ingeniería, Av. San Pablo No. 180, CDMX 02200, Mexico; 2Centro Universitario UAEM Valle de Mexico, Boulevard Universitario S/N Valle Escondido, Río San Javier, Cd López Mateos 54500, Mexico; 3Instituto Politécnico Nacional, CIITEC IPN, Cerrada de Cecati S/N Col. Santa Catarina, Azcapotzalco, CDMX 02250, Mexico; 4Instituto Politécnico Nacional, ESIQIE IPN, Av. Luis Enrique Erro S/N, Nueva Industrial Vallejo, Gustavo A. Madero, CDMX 07738, Mexico

**Keywords:** lutetium vanadate, bismuth-europium co-doping, Pluronic F-127

## Abstract

Nowadays, orthovanadates are studied because of their unique properties for optoelectronic applications. In this work, the LuVO_4_:Eu^3+^, Bi^3+^ films were prepared by the sol–gel method, using a new simple route, and deposited by the dip-coating technique. The obtained films are transparent, fracture-free, and homogenous. The sol–gel process was monitored by Fourier-transform infrared spectroscopy (FTIR), and according to X-ray diffraction (XRD) results, the crystal structure was tetragonal, and films that were highly oriented along the (200) low-energy direction were obtained. The morphological studies by scanning electron microscopy (SEM) showed uniformly distributed circular agglomerations of rice-like particles with nanometric sizes. The luminescence properties of the films were analyzed using a fixed concentration of 2.5 at. % Eu^3+^ and different concentrations of Bi^3+^ (0.5, 1.0, and 1.5 at. %); all the samples emit in red, and it has been observed that the light yield of Eu^3+^ is enhanced as the Bi^3+^ content increases when the films are excited at 350 nm, which corresponds to the ^1^S_0_→^3^P_1_ transition of Bi^3+^. Therefore, a highly efficient energy transfer mechanism between Bi^3+^ and Eu^3+^ has been observed, reaching up to 71%. Finally, it was established that this energy transfer process occurs via a quadrupole–quadrupole interaction.

## 1. Introduction

Nowadays, phosphor materials with luminescent properties prepared by various techniques are extensively studied because of their great ability to convert UV radiation to the visible spectrum and its multiple applications in different fields, such as integrated photonics [1], optics [2,3], and electronics [4,5]. Most luminescent materials are usually oxides, sulfides, or oxysulfides doped with some transition metal or rare-earth element [6,7,8], but in recent years, some lanthanides known as orthovanadates are currently used as matrices due to their high emission efficiency [9,10,11,12]. Additionally, these materials have unique properties due to the stoichiometric combination of their components and their symmetry with the crystal of the complex oxides; these characteristics increase the luminescent activity in comparison with the simple materials [13,14,15,16,17,18]. Likewise, rare-earth ions that are used as impurities in orthovanadates have shown strong emissions as well as variations in the emission color corresponding to the different activating ions under the excitation of an electron beam or ultraviolet. The luminescent properties presented in these materials depend largely on the location of the 4f energy levels of the dopants and the ratio of the valence and conduction bands (VB and CB, respectively) of the matrix [3,12,19,20,21]. In the energy transfer, the co-doped materials are interesting for optoelectronic applications because of increases in their luminescent effect. Particularly for the Eu^3+^ emission, several ions have been proposed to increase its well-known reddish emission, for example, usually by the use of other rare earths, such as Dy^3+^ [22], Tb^3+^ [23], Tm^3+^ [24], or Sm^3+^ [25]. However, the search of other non-rare earths sensitizers has attracted attention in order not only to increase the light yield, but also to modify the excitation wavelength. Therefore, other ions have been studied, like Sb^3+^ [26] or Li^+^ [27]. Another possibility is the use of Bi^3+^, which increases the luminescent properties and broadens the excitation spectrum of europium ions in LuVO_4_ [28] or Lu_2_O_3_ [29]. In the case of LuVO_4_, since it has been shown to have better optical properties than yttrium vanadate [2,8,30], several routes of synthesis have been studied to obtain higher optical properties. However, most of the synthesis routes that have been developed require high temperatures and highly controlled environments to carry out the reaction, for example, in solid state or hydrothermal reaction synthesis. Consequently, new energy-saving simple-synthesis processes are required that can be scalable to industry. In the case of sol–gel synthesis, it allows one to prepare homogeneous, high-purity materials with the possibility of controlling morphologies and impurities at low preparation temperatures without extremely controlled environments. On the other hand, Pluronic F-127 acid was added to the synthesis because it changes the properties of the film, as was shown in Tsotetsi et al. in 2022 [31] and Arconada et al. in 2010 [32], where it was used to increase the thickness because of its viscosity, for providing porosity to the film, and as a morphology controller. Another characteristic of Pluronic F-127 is that it acts as a structure-directing agent in TiO_2_ in catalytic and solar cell applications [33,34]. In the present work, transparent films of the LuVO_4_:Eu^3+^, Bi^3+^ phosphor were obtained using sol–gel synthesis and the dip-coating method, which can be used in white light emission LEDs, using Bi^3+^ as a sensitizer of Eu^3+^ to improve red emission intensity under UV radiation. The films were characterized in terms of their crystalline structure and excitation/emission intensity spectra, as well as in relation to the effect of the annealing temperature and as a function of the Bi^3+^ content.

## 2. Materials and Methods 

### Synthesis

LuVO_4_:Eu^3+^, Bi^3+^ films were prepared using the sol–gel method and dip-coating technique. First, lutetium acetate (CH_3_CO_2_)_3_Lu•xH_2_O (99.9%, Sigma-Aldrich, Saint Louis, MO, USA) was dissolved in ethanol (99.5, C_2_H_6_O, Fermont, Monterrey, México) under vigorous magnetic stirring at 60 °C, obtaining a sol with a 0.15 M lutetium concentration. Later, a second sol was prepared from ammonium metavanadate NH_4_VO_3_ (99.0%, Fermont, Monterrey, México) dissolved in an ammonium hydroxide NH_4_OH (30%, Fermont, Monterrey, México) solution in distilled water (36 M), obtaining a sol with a 0.09 M vanadium concentration after vigorous magnetic stirring at 80 °C for 1 h. Later, the two sols were combined to obtain the LuVO_4_ sol-precursor, and nitric acid was added (0.5 M) to obtain a clear, homogeneous, and bluish solution after 2 h of stirring at 80 °C. On the other hand, to add the Eu^3+^ and Bi^3+^ ions to the lutetium vanadate sol, europium nitrate Eu(NO_3_)_3_·5H_2_O (99.5%, Alfa Aesar, Tewksbury, MA, USA) was incorporated in order to obtain 2.5 mol % Eu^3+^ samples, and bismuth was incorporated from a solution prepared by the dissolution of bismuth nitrate Bi(NO_3_)_3_·5H_2_O (98%, Sigma-Aldrich, Saint Louis, MO, USA) at a 1:1 molar ratio with ethanol:diethyleneglycol (C_4_H_10_O_3_, 98%, Sigma-Aldrich, Saint Louis, MO, USA), obtaining a Bi^3+^ 0.05 M concentration. Later, a fixed volume was incorporated into the lutetium vanadate sol to obtain a Bi^3+^ 0.5, 1, and 1.5 at. % doping-level. Finally, to modify the viscosity of the previously obtained sol and therefore obtain homogeneous films, diethylene glycol (HOCH_2_CH_2_)_2_O (99% Sigma-Aldrich, Saint Louis, MO, USA) was first added (5.7 M), followed by acetylacetone (6.8 M) CH_3_COCH_2_COCH_3_ (99% Sigma-Aldrich, Saint Louis, MO, USA), 0.8 M of HNO_3_, and 0.006 M Pluronic F-127 (C_3_H_6_O·C_2_H_4_O, 99% Sigma-Aldrich, Saint Louis, MO, USA). The final sol can be used to obtain powders or films; for powders, the sol was dried at 100 °C for 24 h and then annealed at several temperatures from 200 to 1000 °C for 3 h for each temperature.

For the dip-coating procedure, the sols were filtered using a 0.2 µm filter, and carefully cleaned silica glass substrates (QSI quartz, Quartz Scientific Inc., Lake County, OH, USA, refractive index = 1.417) were dipped into the prepared sol and pulled up at a constant rate of 4 cm·s^−1^. After each dipping, the films were dried at 100 °C for 10 min to remove the water and the most volatile organics components. Subsequently, the film was annealed at 300 °C and 500 °C for 10 min each to remove the organic remnants. The dipping cycle was repeated 3 times. Finally, to crystallize the LuVO_4_ phase, the films were annealed from 600 to 1000°C for 3 h.

The Fourier-transform infrared spectroscopy (FTIR) spectra of the samples were recorded in the range of 4000–400 cm^−1^, using FTIR (FTIR 2000, Perkin Elmer, Waltham, MA, USA) and the KBr pelleting technique. The phase composition of the powders was identified by X-ray diffraction (XRD) at room temperature on a powder diffractometer (Bruker D8Advance, Billerica, MA, USA), using Cu Kα radiation (1.5418 Å). The morphology studies were carried out with a Zeiss Supra 55VP scanning electron microscope. The luminescence study was carried out at room temperature with a fluorescence spectrophotometer Hitachi F-7000, equipped with a 150 W xenon lamp and an R955 photomultiplier tube.

## 3. Results and Discussion

### Chemical Evolution and Structural Properties

To observe the xerogel chemical evolution during the annealing process in the sol–gel process for the synthesis of LuVO_4_: 2.5% at. Eu^3+^ and 1.5% at. Bi^3+^ sample, the FTIR study at different temperatures (from 200 to 1000 °C) was carried out (Figure 1). From 200 to 400 °C, strong bands at 3300 cm^−1^ (ν), 1650 cm^−1^ (δ), and 750 cm^−1^ (δ) are related to the O-H groups due to the presence of water and alcohols (ethanol and diethilenglycol). All these bands disappear at 600 °C, which indicates that this temperature is adequate to eliminate the O-H groups present in the samples.

In addition, bands associated with the carbonyl groups (-COOH) ~1725–1700 cm^−1^ are observed and attributed to the stretching bond of the lutetium acetate precursor employed, and are completely removed at 600 °C, which is an indicator of the moment of the beginning of crystallization process [8]. Bands around ~1430–1100 cm^−1^ are observed, which correspond to the symmetric and asymmetric vibrations of the C-H bonds, probably from the use of Pluronic F-127; however, as can be observed, they disappear at 700 °C [35,36]. Next, at 500 °C, bands related to the crystallization of LuVO_4_ can be observed at ~810–830 cm^−1^ and ~446–455 cm^−1^, which correspond to the V-O bonds of the vanadate group (VO_2_^3−^) and Lu-O, respectively [2]. It is important to notice that, from 700 °C, all the organic groups have been completely removed from the LuVO_4_ system, and, therefore, all the luminescent properties correspond only to the ceramic film. The same results have been obtained for the 0, 0.5, and 1.0 at. % of Bi^3+^ (not shown). All the previous observations can be verified by the diffraction patterns obtained by XRD.

First, the XRD patterns of LuVO_4_: 2.5 at. % Eu^3+^, X at. % Bi^3+^ (X = 0.5, 1, 1.5) powders derived from the same sol used to prepare the films, annealed at 1000 °C, are presented in Figure 2. It can be observed that all samples are exactly in agreement with the corresponding 98-041-9281 ICSD from LuVO_4_, and the only appreciable impurity, observed at ≈ 26.05°, could be related to the formation of a secondary vanadate phase, LuVO_3_ (98-018-5830 ICSD). It is important to notice that no evidence of the possible formation of impurities from Bi^3+^ or Eu^3+^ ions has been detected. This result is expected for Eu^3+^ because, being a rare earth, it can occupy the position of Lu in the vanadate structure. On the other hand, regarding the case of bismuth, since it would also present a tetragonal structure forming BiVO_4_ and the same valence as Lu, it can be possible to have at least a partial dissolution of Bi in the structure. Furthermore, the inset in Figure 2 shows that the principal LuVO_4_ peak presents a gradual shift to the lower degree with the increase in Bi^3+^ content, implying the lattice expansion of LuVO_4_ when the Lu atoms in the lattices are gradually substituted by Bi, which has a bigger radius. Similar results have been reported previously by for the YVO_4_ system [37], who reported the appearance of the BiVO_4_ phase until the 4.0 at. % of Bi^3+^. Lu M. et al. [38]. also reported the absence of impurities for YVO_4_: 8% Eu, 10% Bi nanoparticles Finally, Zeng L. et al. [39] also reported the substitution of Lu^3+^ by Bi^3+^ ions in nanoparticles of LuVO_4_ obtained by a hydrothermal method.

On the other hand, Figure 3 shows the structural evolution of the LuVO_4_: 2.5% at. Eu^3+^, 1.5% at. Bi^3+^ film as the annealing temperature increases. As observed, the crystallization process begins at 650 °C, which confirms the FTIR observation. The diffraction peaks can be ascribed to the diffraction pattern of tetragonal lutetium vanadate LuVO_4_ (98-041-9281, ICSD), space group I41/amd, and with the lattice systems a (7.01 Å), b (7.01 Å), and c (6.19 Å).

It is evident that the increment in the annealing temperature increases the crystallization of the films. At 600–650 °C, the broad peak centered at ~25–25.5° is typical for the amorphous structure obtained by the xerogel formation by the sol–gel process [40]. This amorphous structure disappears at 700 °C and with the increment in temperature, other peaks characteristic of the lutetium vanadate structure do not appear, which indicates that the films are highly oriented towards the (200) direction. In general, this result can probably be attributed to the films which, when deposited, tend to be structured first in the direction of lower surface energy, in this case, the (200) direction. Additionally, the presence of F127 probably tends to decrease the surface energy of the deposited xerogel; thus, the films maintain this preferential orientation. Furthermore, it has been stated [41] that the slow evaporation of high-molecular-weight alcohol, upon heating, like the diethylene glycol used in the present work, allows the structural orientation of the film before crystallization, preferable in the kinetically favored orientation along the c-axis (200). In order to quantify this observation, the preferential orientation parameter α_hkl_ was calculated, which is defined as [42]:(1)$$\propto_{hkl}=\frac{I_{hkl}}{\sum I_{hkl}}$$
where I*_hkl_* is the relative intensity of the corresponding diffraction. Table 1 shows the (200) preferential orientation-index α_200_ of the prepared films, and as observed, it practically does not present changes due to the effect of temperature increase, remaining at approximately 0.90, which shows that the orientation process was carried out from low temperatures and, once the material crystallized, the system tends to maintain that orientation. It is worth mentioning that when Pluronic F-127 acid is used, in some studies, it has been shown that it changes the size of the particles and their orientations [36,37,38,39,40,41,42,43], which is a very desirable property for luminescent films since oriented films tend to minimize the scattering of light during emission process, therefore incrementing its quantum efficiency [44,45]. On the other hand, the crystallite size does not present a significant difference with the increment in the annealing temperature, as can be observed in Table 1. In this regard, the crystallite size was computed by the Scherrer equation:(2)$$D=\frac{K\lambda }{\beta \cos\theta }$$
where D is the crystallite size, *K* is a constant (0.9), λ is the X-ray wavelength (Cu-Kα = 0.15418 nm), β is the full width at half maximum (FWHM), and θ is the half diffraction angle of the peak centroid.

## 4. Morphological Studies

The obtained films were transparent and fracture-free, as can be observed in Figure 4a. Additionally, the luminescent macroscopic behavior under short and large UV- Vis excitation are presented in Figure 4b,c, respectively.

The scanning micrographs of the films, shown in Figure 5, correspond to LuVO_4_: 2.5 at. % Eu^3+^, 1.5 at. % Bi^3+^ film, deposited and annealed at 700 °C (Figure 5a) and 1000 °C (Figure 5b) at 10,000×.

Micrographs of films containing LuVO_4_: 2.5 at. % Eu^3+^, 1.5 at. % Bi^3+^ show a distribution of circular agglomerations of rice-like particles (according to NIST), morphology that is typical of vanadate [46,47,48], and as observed, these particles are distributed over the whole surface. The particle size was measured directly from the micrographs of 180 individual particles for both samples in such a way that it was possible to determine the average size is 579.3 ± 96.7 nm and 716.4 ± 123.8 nm, for the 700 °C and 1000 °C, respectively. The increment in the particle size can be explained since the energy excess with the higher temperature promotes the growth of particles. From Figure 4c, which corresponds to the film annealed at 1000 °C at 500×, it is possible to observe that these particle growth zones are distributed over the surface. Additionally, from the results shown in XRD, it was determined that the films grow preferentially in the (200) direction; therefore, the morphology of the particles corresponds to the growth in this direction. Similar morphologies have been observed in other yttrium vanadate films [49]. Finally, it is important to note that this preferential growth of the films is probably the product of the presence of Pluronic F-127 and diethylene glycol, which, as evaporate, would promote the formation of some pores [50,51].

## 5. Photoluminescence Study

Figure 6 shows the excitation and emission spectra corresponding to the LuVO_4_: 2.5% at. Eu^3+^, 1.5% at. Bi^3+^ films annealed at 1000 °C. The excitation spectrum shows a wide band at 250–380 nm, where the absorption of vanadate groups at λ_em_ = 615 nm can be assigned (VO_4_^3−^) according to Liang et al. [22]. On the other hand, the emission spectrum of the films of the LuVO_4_:Eu^3+^, Bi^3+^ system is shown under excitation at 350 nm, and peaks can be observed at 594, 615, 650, and 700 nm that correspond to the transitions of Eu^3+^ ions: $5D_{0}^{}\to 7F_{J}^{}$ (J = 1, 2, 3, 4), respectively, of which the one located at 615 nm is the most acute, therefore it is the one with the highest emission. The band of bismuth emission, located at 550 nm and generated with 350 nm excitation, decreases, as shown by the curve in the blue color, which has been scaled by ten. The best luminescent properties were reported in the case of the higher bismuth content, which could be explained by the charge transfer of the metal-metal bond (Bi^3+^, V^5+^) and the subsequent energy transfer to Eu^3+^ [20], and a better index of refraction, density, possibly a reduction of the pores, and a structural alignment of the films, which reduces the dispersion of the emissions and for that reason, better patterns are obtained [52].

On the other hand, the luminescence properties of bismuth ions were analyzed. Figure 7a presents the excitation and emission spectra of Bi^3+^ for a mono-doped LuVO_4_: 1.0 at. % Bi^3+^ film. As can be observed, congruent results are obtained when comparing with the observations of the co-doped sample, being the wavelength excitation at 332 nm and the greenish emission at 550 nm, products of the energy transition ^1^S_0_→^3^P_1_ for the excitation process and vice versa for the emission one. However, as also observed in Figure 6, for the co-doped samples, the greenish emission almost disappears, which clearly indicates an energy transfer process from Bi to Eu. Figure 7b presents the reduction of the 550 nm emission from Bi^3+^ in the co-doped samples, compared with the mono-doped one. As the Bi^3+^ content increases, the 550 nm emission almost disappears because of the energy transfer.

Therefore, the previous results establish an energy transfer process from Bi^3+^ to Eu^3+^. Similar behavior has been observed by Zhu et al. (2013) [53], and it is possible to propose the mechanism for LuVO_4_:Eu^3+^, Bi^3+^ presented in Figure 8. First, when LuVO_4_ is excited by the UV light, the energy could be absorbed by Bi^3+^ via the ^1^S_0_→^3^P_1_ transition. Next, there are two different types of excited states of Bi^3+^, ^3^P_1_, which relaxes to the ^1^S_0_ ground state, producing a broad luminescence band that could transfer energy to the Eu^3+^. Thereafter, the absorbed energy can be transferred to the ^5^D_2_ level of Eu^3+^, and the emission of Bi^3+^ partly quenches. At the same time, Eu^3+^ ions can also excite to ^5^D_2_ from the ground state, ^7^F_0_, and all the energy falls to the ground state, ^5^D_0_. Immediately, red light emission peaks at 615 nm occur because of the ^5^D_0_→F_2_ characteristic transitions of Eu^3+^ ions.

The influence of the Bi^3+^ content on the luminescence of Eu^3+^ is observed in Figure 9a at λ_exc_ = 350 nm. In this case, the intensity at 615 nm increases when the atomic content of bismuth is bigger, while the emission observed at 538 nm (corresponding with the bismuth emission) reduces in ratio with the increase in europium emission because an energy transfer occurs according to the energy diagram (Figure 8). For comparison purposes, the emission spectrum of an Eu-mono-doped sample has been added, with λ_exc_ = 308 nm, to observe that there is indeed a process of increment in the emission of Eu^3+^ due to the presence of bismuth. Furthermore, the emission of Eu^3+^ is highly dependent on the surroundings of the cation. It is possible to set the ratio R = I (^5^D_0_ →^7^F_2_)/I (^5^D_0_ →^7^F_1_) as a reference to the coordination state and symmetry since the intensity of the ^5^D_0_ →^7^F_1_ magnetic dipole transition (centered at 594 nm) is independent of the surroundings of the Eu^3+^, whereas the intensity of the ^5^D_0_ →^7^F_2_ electric dipole transition (centered at 618 nm) is very sensitive to site symmetry. If the R value is low, the Eu^3+^ cation tends to localize at a centrosymmetric site or a high-symmetry site. In contrast, when the R value is high, the Eu^3+^ cation tends to localize at a non-centrosymmetric site or a low-symmetry site [54]. For the LuVO_4_: 2.5% at. Eu^3+^, X% at. Bi^3+^, the R factor increases from 3.91→4.73→4.81 for X = 0, 0.5, 1, and 1.5, respectively, showing that the increment in the Bi^3+^ content promotes the emission of the low-symmetry sites.

On the other hand, it is possible to calculate the energy transfer efficiency (ET) between Bi^3+^→Eu^3+^ [55]:(3)$${ŋ}_{ET}=1-\frac{I_{S}}{I_{S0}}$$
where *I*_𝑆_ is the luminous intensity of Bi^3+^ ions in the presence of Eu^3+^ ions, and *I*_𝑆0_ is the luminous intensity of Bi^3+^ ions in the absence of Eu^3+^ ions. The energy transfer efficiency, as observed in Figure 9b, shows that as the Bi^3+^ content increases, the energy transfer efficiency from Bi^3+^→Eu^3+^ also increases, reaching high values up to 71% in the 1.5 at. % sample. The ET values vary depending on the matrix and the co-doped ion. For comparison, for other proposed sensitizers for Eu^3+^, an ET of 68% was obtained for Tm^3+^→Eu^3+^ in LiLaSiO_4_:Tm^3+^, Eu^3+^ [56]; 78% for Tb^3+^→Eu^3+^ in Ba_2_SiO_4_ [57]; or 17% for Sm^3+^→Eu^3+^ on TeO_2−_GeO_2−_ZnO glasses [24].

On the other hand, the energy transfer from the sensitizers to the activators could be achieved through electric multipole interactions. According to the Dexter energy level transfer, the electric multipole interactions can be related to [3]:(4)$$\frac{I_{S}}{I_{S0}}\approx \frac{{ŋ}_{0}}{ŋ}\;\propto C^{n/3}$$
where *ŋ_0_* and *ŋ* represent the quantum efficiencies of Bi^3+^ ions in the absence and presence of Eu^3+^ ions, respectively, the ratio 𝜂_0_/𝜂 can be calculated by the ratio of luminous intensities (*I*_𝑆0_/*I*_S_), and *C* is the total molar concentration of Bi^3+^ and Eu^3+^, whereas *n* = 6, 8, or 10 correspond to electric dipole–electric dipole, electric dipole–electric quadrupole, or electric quadrupole–electric quadrupole interactions, respectively. The corresponding plots are presented in Figure 10, and as observed, the best linear fit is exhibited when *n* = 10/3, implying that the most probable interaction for the energy transfer from Bi^3+^ to Eu^3+^ in the LuVO_4_ system occurs via quadrupole–quadrupole interactions. This electronic interaction is in good agreement with other Bi^3+^→Eu^3+^ energy transfer processes on other systems; for example, the same process has been observed in a Bi^3+^/Eu^3+^-doped SiO_2_–Al_2_O_3_–CaO–B_2_O_3_–La_2_O_3_–K_2_O glass, with an ET efficiency of 86% [58], and also on the Ca_10_(PO_4_)_6_F_2_:Bi^3+^, Eu^3+^, with an ET efficiency of 50% [53].

To corroborate the energy-transfer process, the Bi^3+^ fluorescence decay behavior has been studied. Figure 11 shows the fluorescence decay curves of LuVO_4_: Eu^3+^, Bi^3+^ films at 550 nm when the systems are excited at 350 nm. The best fit of the experimental data was obtained using the double exponential equation [59]:(5)I(t)=I0+A1e−tτ1+A2e−tτ2
where *I*(*t*) is the luminescence intensity at time *t*, *τ*_1_ and *τ*_2_ are rapid and slow decay times for the exponential components, respectively, and *A*_1_ and *A*_2_ are constants.

That the double exponential equation presented the best fit can be explained since, from classical kinetics, there are two emitting species [60], which is the case in co-doped systems. S Dutta et al. [61] established that this can occur for systems with low concentrations of lanthanides and that it can be due to three factors: (1) the difference in the nonradiative probability of decays for lanthanide ions at or near the surface and lanthanide ions in the core; (2) an inhomogeneous distribution of the doping ions in the host material, leading to the variation in the local concentration; and (3) the transfer of excitation energy from the donor to lanthanide activators [62,63,64].

From the calculation of the decay values of τ_1_ and τ_2_, it can be observed that the values of the co-doped samples are smaller compared with the mono-doped Bi^3+^ sample, which corroborates the energy mechanism process between Bi^3+^ ions to Eu^3+^ ions. Furthermore, as the relation of Eu^3+^/Bi^3+^ increases (from 1.5 to 0.5 Bi^3+^ samples), the fluorescence lifetime decreases. This effect can be explained by the fact that, as this Eu^3+^/Bi^3+^ ratio increases, the Bi^3+^→Eu^3+^ ion distance must shorten, and the interaction gradually increases, so the fluorescence lifetime of Bi^3+^ ions decreases. The fast decay component (τ_1_) is derived from the fluorescence decay, whereas the slow decay component (τ_2_) possibly results from the luminescence caused by defects of phosphors on the surface and in the bulk structure [65].

On the other hand, the quality of the light emission is evaluated by the CIE 1931 color matching function, the correlated color temperature (CCT), and the color purity (CP), to understand the fluoresce center [66,67,68,69]. Table 2 presents the CIE color coordinates for different concentrations of bismuth; the mono-doped Bi film presents coordinates (0.3611, 0.5124) that are congruent with the results of Krishnan (2018), which studied bismuth silicate glasses for white light generation and bismuth silicate oxyfluoride glasses for LED applications. The calculated CCT values are below 3000 K in the co-doped films and, therefore, could be useful for warm light sources. When Bi is incorporated with 2.5 at. % Eu, the CP improved to 0.86. This highest CP and low CCT with the CIE coordinates moving to the red emission region is due to the energy transfer, as explained earlier, which is congruent with similar reports [70,71].

Table 2 also shows the RGB coordinates, calculated from the x and y coordinate values, and the hex values, from the RGB coordinates [72,73]. The hex code represents a useful tool to present the actual color display used in LED displays. As observed, all the co-doped samples present reddish color, whereas the Bi mono-doped sample has a greenish color.

Finally, the CIE chromaticity coordinates (Figure 12) proved that when the bismuth concentration is 1.5 at. %, the color of the emission is reddish, according to the diagram. A displacement from emitting a yellow-green color to a red color appears when the pure bismuth (100%) is combined with a constant europium concentration (2.5 at, %); however, it is evident that the best energy transfer occurs with a greater amount of bismuth because in the case of 0.5 and 1% europium the displacement is minimal between them, but emission changes to the color red. This result is congruent with that obtained by Zhang (2018) [28].

## 6. Conclusions

A new sol–gel synthesis was carried out to easily obtain films of the LuVO_4_ system at different concentrations of bismuth, LuVO_4_: 2.5% at. Eu^3+^, X% at. Bi^3+^ (X = 0, 0.5, 1, and 1.5), using the dip-coating method at low temperatures. The results show that the first bond of Lu-O is obtained from 500 °C, and it is possible to obtain completely structured films from 700 °C. Furthermore, it has been stated that, due to the presence of Pluronic F-123 and diethylene glycol during the sol–gel process, it has been possible to obtain highly oriented films along the (200) direction. Luminescence studies show that there is an energy transfer mechanism from Bi^3+^ to Eu^3+^, which effectively increases the light yield of the ^5^D_0_→^7^F_2_ transition of Eu^3+^. Furthermore, it was shown that the energy transfer increases as the Bi^3+^ concentration increases, reaching an energy transfer efficiency for Bi^3+^→Eu^3+^ of 71% with 1.5 at. % Bi^3+^_,_ and it was observed that the energy transfer takes place via a quadrupole–quadrupole interaction. Finally, the actual real color from the co-doped samples is reddish, which is typical for the presence of Eu^3+^. The produced films could have different possible applications in white emission LEDs, mobile devices screens, lasers, and optoelectronic devices.

## Figures and Tables

**Figure 1 materials-16-00146-f001:**
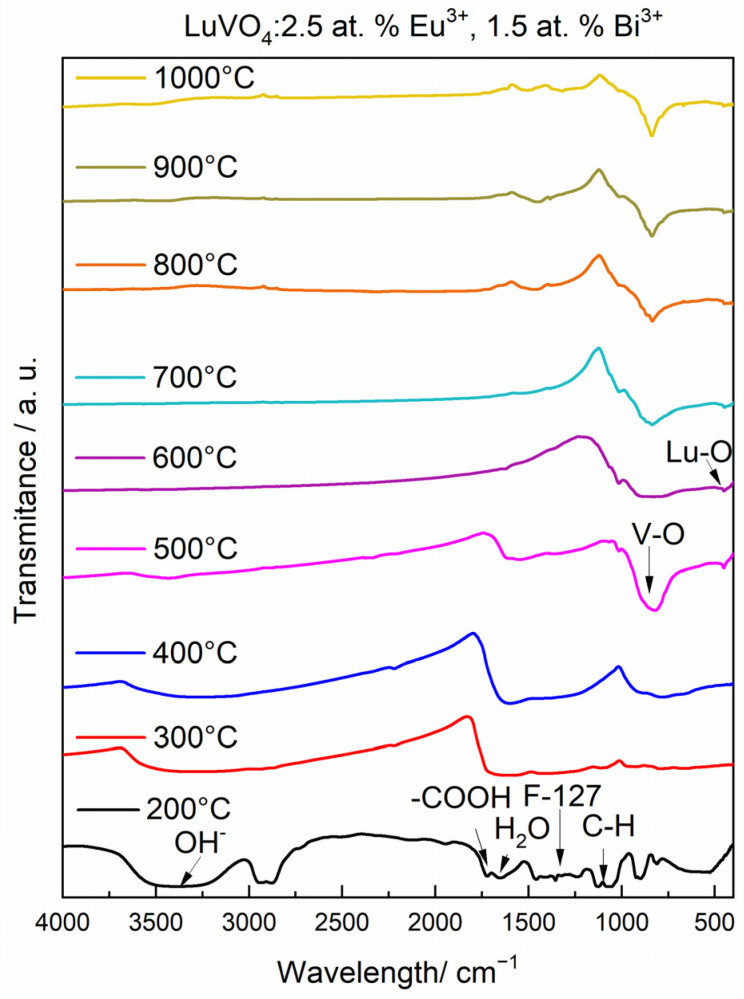
FTIR spectra of LuVO_4_: 2.5% at. Eu^3+^, 1.5% at. Bi^3+^ chemical evolution.

**Figure 2 materials-16-00146-f002:**
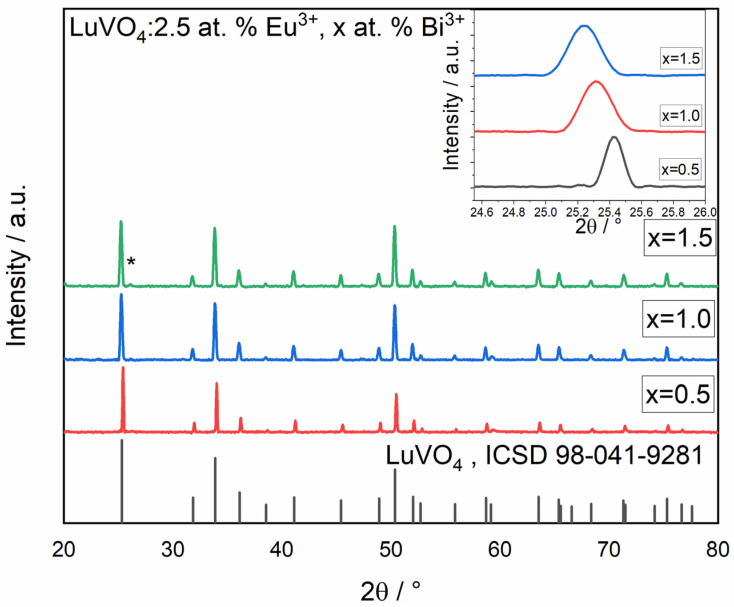
The XRD patterns of LuVO_4_: 2.5 at. % Eu^3+^, X at. % Bi^3+^ (X = 0.5, 1, 1.5) powders prepared from the same sol of the films. The inset is the partial magnification of the (200) peak. On X= 1.5 sample, at ≈ 26.05° (*) the secondary phase LuVO_3_ it’s observed.

**Figure 3 materials-16-00146-f003:**
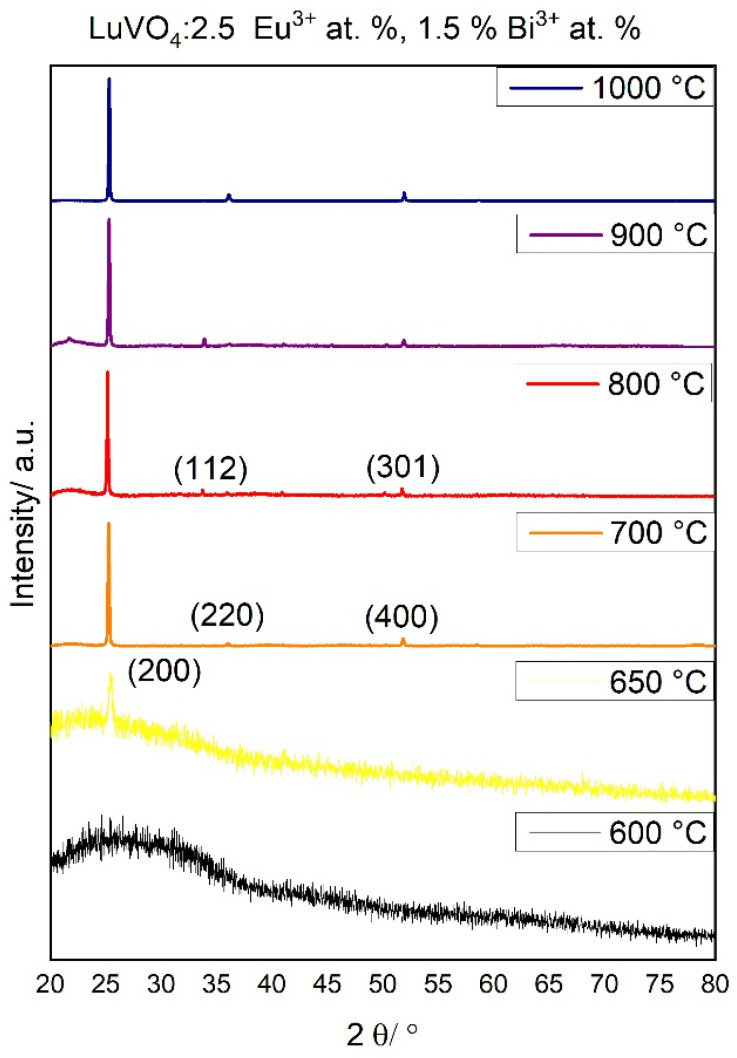
Structural evolution of LuVO_4_: 2.5% at. Eu^3+^, 1.5% at. Bi^3+^ films by XRD analysis.

**Figure 4 materials-16-00146-f004:**
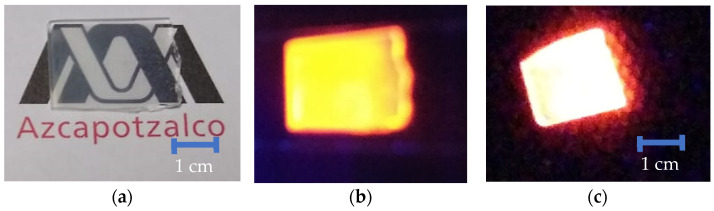
Photography of LuVO_4_: 2.5% at. Eu^3+^, 1.5% at. Bi^3+^ films (**a**) without, (**b**) with UV-Vis short excitation (254 nm), and (**c**) with long UV-Vis on excitation (380 nm).

**Figure 5 materials-16-00146-f005:**
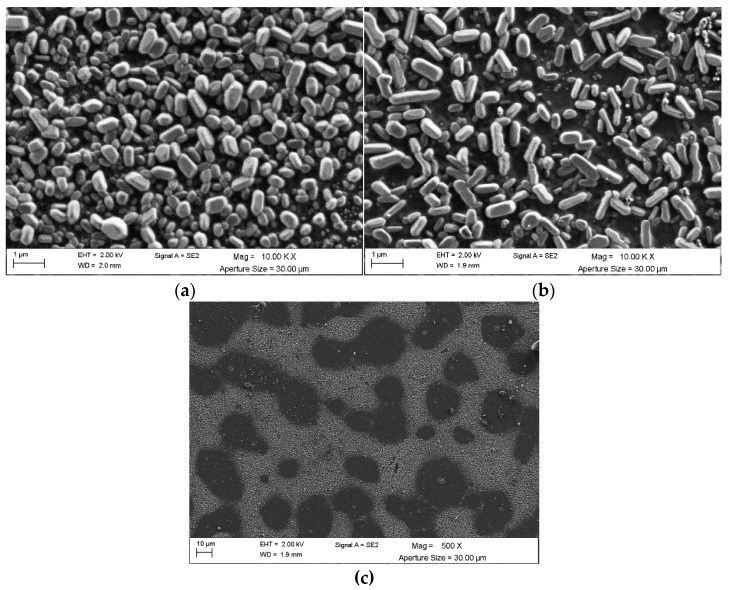
SEM micrographs of LuVO_4_: 2.5% at. Eu^3+^, 1.5% at. Bi^3+^ films films: (**a**) after annealing for 3 h at 700 °C, 10,000×, (**b**) after annealing at 1000 °C for 3 h, 10,000×, and (**c**) after annealing at 1000 °C for 3 h, 500×.

**Figure 6 materials-16-00146-f006:**
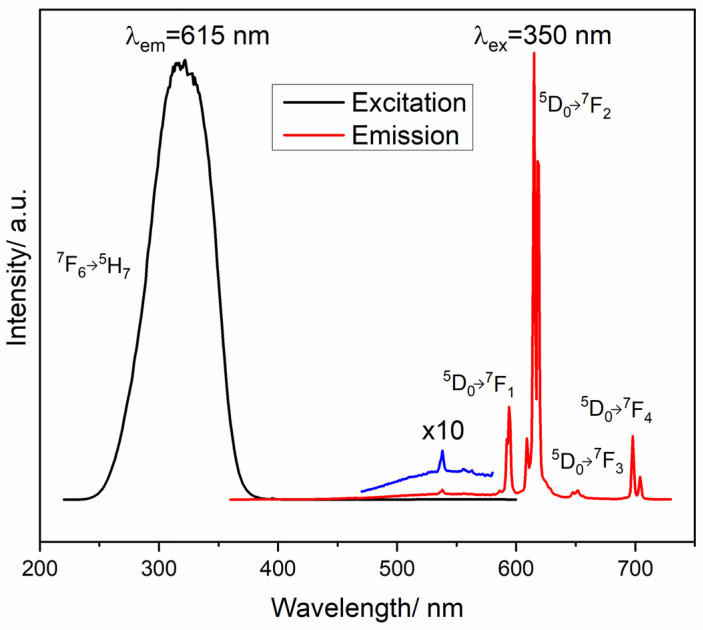
Excitation and emission spectra of LuVO_4_ films with 2.5 at. % Eu^3+^ and 1.5 at. % Bi^3+^ after 3 h of annealing at 1000 °C.

**Figure 7 materials-16-00146-f007:**
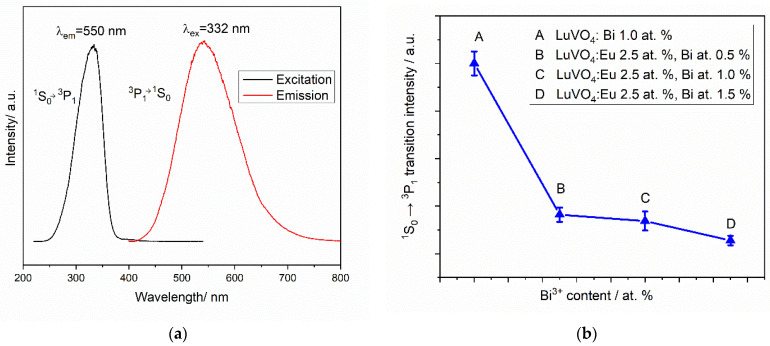
(**a**) Excitation and emission spectra of Bi^3+^ and (**b**) Intensity of 550 nm Bi^3+ 1^S_0_→
^3^P_1_ transition.

**Figure 8 materials-16-00146-f008:**
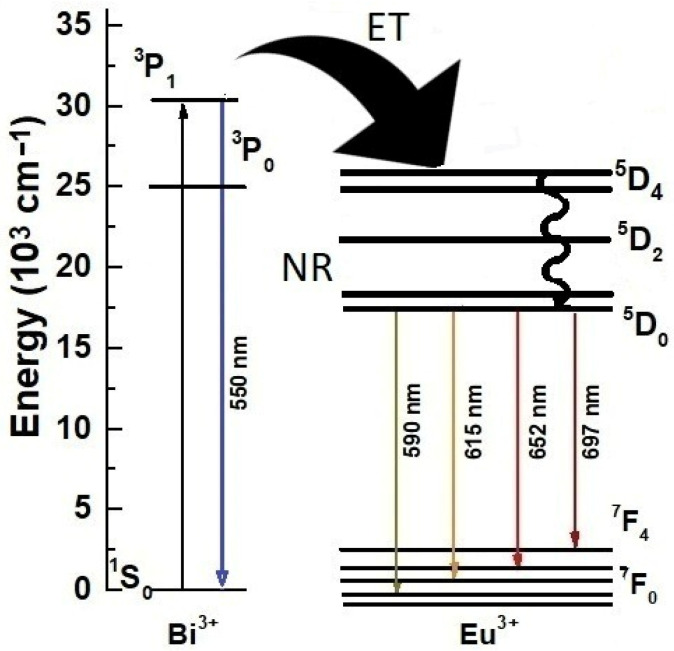
Energy level diagram of Bi^3+^ and Eu^3+^ in LuVO_4_.

**Figure 9 materials-16-00146-f009:**
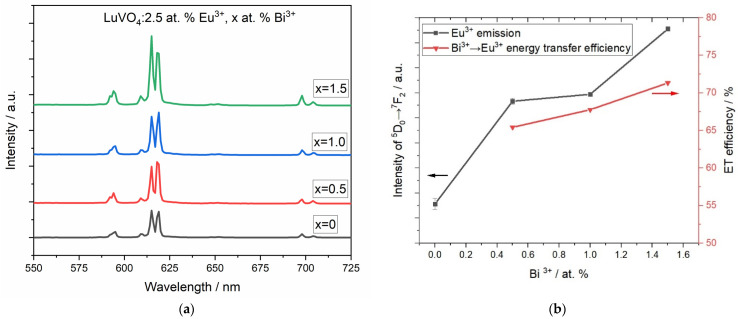
(**a**) Emission spectra of LuVO_4_: 2.5 at. % Eu^3+^, X Bi at. % (X = 0.5, 1.0 and 1.5) at λ_exc_ = 350 nm. For the mono-doped sample, λ_exc_ = 302 nm for comparison purposes. (**b**) Influence of the Bi^3+^ content on the luminescence of Eu^3+^ and energy transfer efficiency of Bi^3+^→Eu^3+^.

**Figure 10 materials-16-00146-f010:**
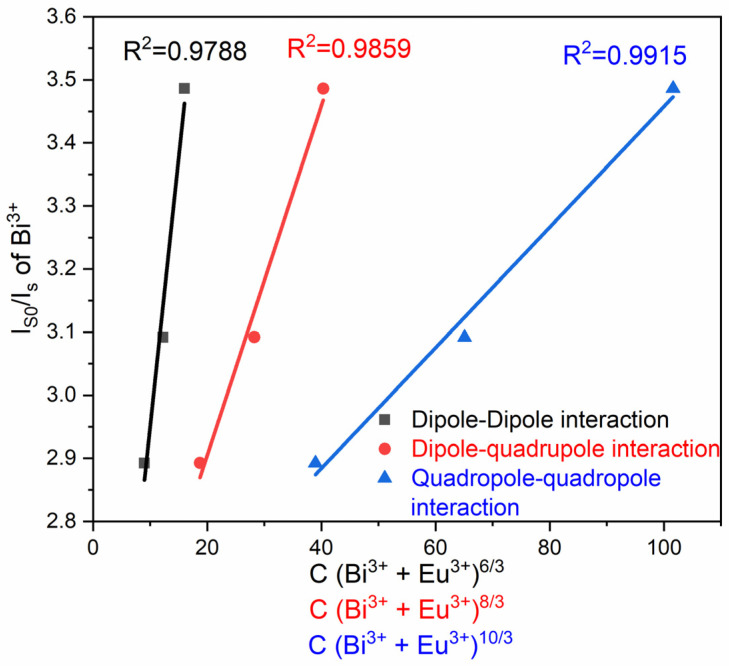
Fitting relationship of I_S0_/I_S_ of Bi^3+^ on C(Bi^3+^+Eu^3+^)^6/3^ for a dipole–dipole interaction; C (Bi^3+^+Eu^3+^)^8/3^ for a dipole–quadrupole interaction, and C(Bi^3+^+Eu^3+^)^10/3^ for a quadrupole–quadrupole interaction.

**Figure 11 materials-16-00146-f011:**
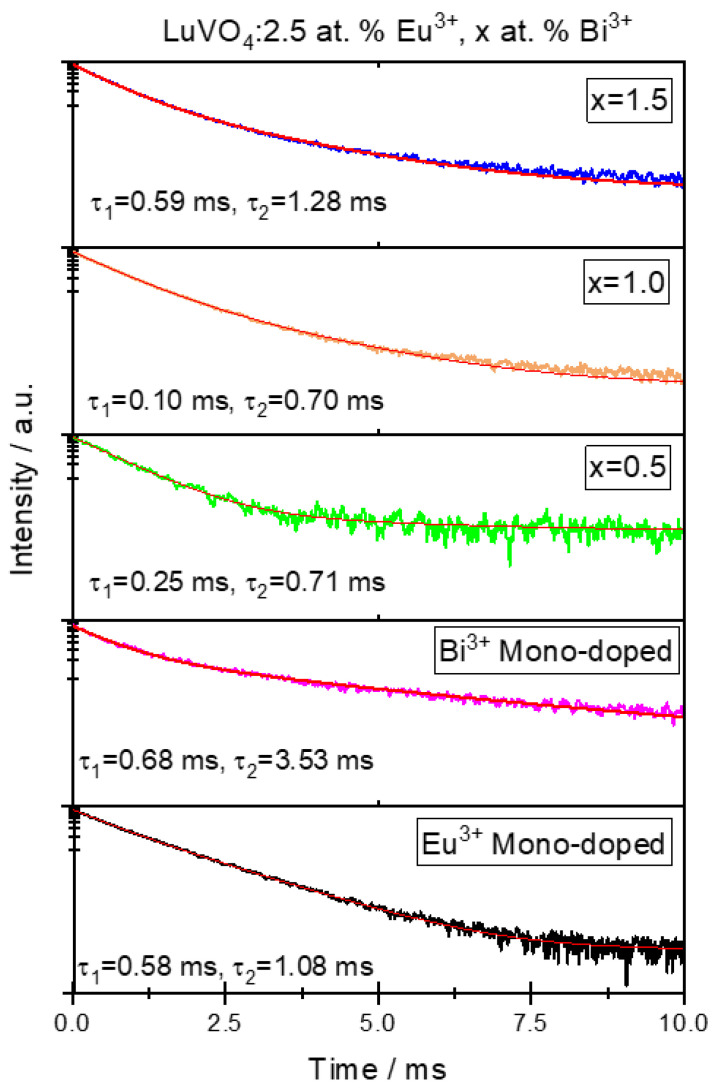
Decay curves of LuVO_4_:Eu^3+^, Bi^3+^ films.

**Figure 12 materials-16-00146-f012:**
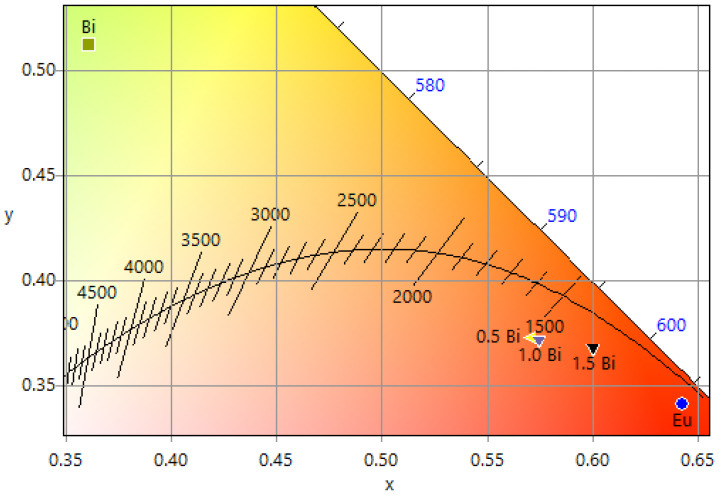
CIE chromaticity coordinates for the LuVO_4_: 2.5 at. % Eu^3+^, X at. % Bi^3+^ (X = 0.5, 1, 1.5) films and a Bi^3+^ mono-doped sample.

**Table 1 materials-16-00146-t001:** Crystallite size and preferential orientation index of LuVO_4_: 2.5% at. Eu^3+^, 1.5% at. Bi^3+^ films as a function of annealing temperature.

System	Annealing Temperature (°C)	Crystallite Size (nm)	Preferential Orientation Index α_200_
LuVO_4_: Eu^3+^ 2.5 at. %, Bi^3+^ 1.5 at. %	700	68.2 ± 6.2	0.92
	800	71.6 ± 7.8	0.91
	900	66.9 ± 4.2	0.91
	1000	80.8 ± 6.1	0.90

**Table 2 materials-16-00146-t002:** Chromaticity coordinates (x, y), correlated color temperature (CCT), color purity (CP), RGB coordinates, and Hex color.

Sample	CIEE Color Coordinates (x, y)	CTTK	CP	RGB			Hex	Hex Color
				R	G	B		
Eu^3+^ mono-doped	(0.6423, 0.3416)	1026	0.92	255	34	0	#FF2200	
Bi^3+^ mono-doped	(0.3611, 0.5124)	4939	0.59	189	255	80	#BDFF50	
Eu^3+^ 2.5 at. %, Bi^3+^ 0.5 at. %	(0.5713, 0.3734)	1620	0.78	255	97	33	#FF6121	
Eu^3+^ 2.5 at. %, Bi^3+^ 1.0 at. %	(0.5721, 0.3724)	1620	0.79	255	96	33	#FF0021	
Eu^3+^ 2.5 at. %, Bi^3+^ 1.5 at. %	(0.5999, 0.3684)	1680	0.86	255	81	0	#FF5100	

## Data Availability

Not applicable.

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
