# Peer review of "Luminescence Properties and Energy Transfer of Eu3+, Bi3+ Co-Doped LuVO4 Films Modified with Pluronic F-127 Obtained by Sol–Gel"

_materials, 2022, doi:10.3390/ma16010146_

Round 1
Reviewer 1 Report
The article concerns the study of the luminescence and energy transfer properties in materials based on LuVO4 co-doped with Bi3+ and Eu3+ and modified with F-127 Pluronic, obtained with the sol-gel method.
From a scientific point of view, in general, the article is quite interesting. Unfortunately, the English writing is very difficult to understand, and this does not allow to fully understand the study carried out by the authors.
In the "Introduction", more information on co-doped materials in the literature would be needed.
As regards the "Methodology" section, the part relating to the methods and analysis instruments used is missing. The synthesis descriptions are also not described in an understandable way, unfortunately.
Regarding the "Results", in general the data should be described in more detail. A comparison should be made with reference materials from the literature and/or with a material based only on LuVO4:Eu3+ (which is missing from the article).
"Conclusions" should be written in more detail by including the data presented in the article.
As a general note, an extensive revision by the authors is needed to further evaluate the publication of the article on the materials.
Questions:
1) Did the authors perform UV-Vis analyses on the samples produced (in addition to the images in Figure 3)?
2) Compared to parent materials in the literature, how are the samples produced in terms of performance?
3) In the photoluminescence studies, did the authors determine the asymmetry factor for europium centres in the samples produced? Regarding the energy transfer, what are the values ​​of the energy and rate parameters (if calculated)? Have europium's coordinated water molecules been evaluated?
4) What type of Colour Space was used for the analysis of the photometric properties? Did the authors also calculate the relative RGB and Hex values ​​from the chromaticity coordinates?
Author Response
The authors wish to acknowledge the kind comments of the reviewer and the time invested in the review, which greatly improved this manuscript.
The article concerns the study of the luminescence and energy transfer properties in materials based on LuVO4 co-doped with Bi3+ and Eu3+ and modified with F-127 Pluronic, obtained with the sol-gel method.
From a scientific point of view, in general, the article is quite interesting. Unfortunately, the English writing is very difficult to understand, and this does not allow to fully understand the study carried out by the authors.
In the "Introduction", more information on co-doped materials in the literature would be needed.
R= More information has been added, including 6 new references.
As regards the "Methodology" section, the part relating to the methods and analysis instruments used is missing. The synthesis descriptions are also not described in an understandable way, unfortunately.
R= The entire experimental section has been rewritten, in such a way that it can more clearly represent the new proposed sol-gel process, in such a way that it can be reproduced.
Regarding the "Results", in general the data should be described in more detail. A comparison should be made with reference materials from the literature and/or with a material based only on LuVO4:Eu3+ (which is missing from the article).
R= A large part of the results have been rewritten, with the aim of having a better understanding of the phenomenon of energy transfer from bismuth to europium, for which, the following were added: calculations of energy transfer efficiency, study of decay times, studies related to the knowledge of the mechanism that controls the transfer of energy. Likewise, new results were added in the structural characterization part related to the evolution of the structure as a function of the bismuth content. In general, the discussion of results was increased.
On the other hand, with the new calculations of the energy transfer efficiency, as well as the calculation of the mechanism by which this is carried out, comparisons are made with respect to other works to observe the energy transfer efficiency of Bi a Eu presented in this work.
Regarding the europium monodoped sample, the corresponding emission spectrum was added, as well as the CIE, RGB, and Hex coordinates, for comparison.
R= The emission spectrum of the Europium monodoped sample was added (Figure 9a, x=0), which clarifies that the excitation at 302 nm was taken for this spectrum, since at 350 nm, which is the excitation used in the co-doped samples, there is no significant emission. It should be remembered that the study contemplates the transfer of energy from Bi to Eu, which was shown to have its maximum at 350 nm (Figure 6). Additionally, the x, y, RGB and Hex values ​​of the sample were added to table 2 for comparison.
Added RGB and Hex values ​​in table 2, additionally added RGB "real" color
"Conclusions" should be written in more detail by including the data presented in the article.
R= The conclusions section has been rewritten and the requested results have been added.
As a general note, an extensive revision by the authors is needed to further evaluate the publication of the article on the materials.
R= It will be noted that much of the manuscript was rewritten, new results added, introduction modified, methodology and conclusions completely rewritten. In addition, all the grammar was revised, titles and subtitles were homogenized, etc.
Questions:
- Did the authors perform UV-Vis analyses on the samples produced (in addition to the images in Figure 3)?
R= Two new images of the luminescence of the films at short and long wavelengths (254 nm and 380 nm) were added, the first being typical for conventional Eu emission, and the second the closest to the maximum found in the present work,
- Compared to parent materials in the literature, how are the samples produced in terms of performance?
R= The comparison with other similar materials reported regarding the calculation of the energy transfer efficiency (lines 316-324) as well as the mechanism by which it is carried out (lines 342-346) was added.
- In the photoluminescence studies, did the authors determine the asymmetry factor for europium centres in the samples produced? Regarding the energy transfer, what are the values ​​of the energy and rate parameters (if calculated)? Have europium's coordinated water molecules been evaluated?
R= The R factor that determines the degree of asymmetry related to the luminescence of Europium was calculated, and it was found that the emission of the 5D0-7F2 transition, that is, the most anti-symmetric, is favored by the increase in Bi content. of the sample, which confirms that there is indeed such an energy transfer mechanism. The discussion is shown on lines 292-308. Likewise, the energy transfer efficiency was calculated (lines 311 to 324), where the results are explained, in addition to being added in Figure 9b for better understanding. In general, it was also determined that the increase in bismuth increases the transfer efficiency
Regarding the effect of water molecules, we allow ourselves to mention that, as shown in Figure 2, practically from 700 °C the presence of water is no longer found within the system, since it has been eliminated by the annealing temperature. In fact, from 700 °C, the presence of OH bonds, that could be ascribed to the presence of water is not observed. Additionally, the synthesized crystalline structure, LuVO4 ceramic, does not present coordination water within its structure, and practically is the only phase detected by XRD (Figure 3) For all the above, the luminescent effect of water is not analyzed. Added explanatory note of complete elimination in the FTIR section, lines 136-139.
- What type of Colour Space was used for the analysis of the photometric properties? Did the authors also calculate the relative RGB and Hex values ​​from the chromaticity coordinates?
The CIE 1931 color space was used. The corresponding clarification was added (line 375). Additionally, the RGB coordinates were calculated from the CIE coordinates, as well as the Hex coordinates (lines 384-390 and table 2).
Reviewer 2 Report
- The whole manuscript needs extensive editing of the English language which makes the readers hard to follow.
I can only mention a few examples
Line 68: thermic treatment
Line 72: synthesized sol-gel method
Line 160-161: I think you mean Figures 4a and 4b
Line192: mayor bismuth
- The synthesis part of the methodology section should be revised.
- The best intensity was obtained using the higher concentration of Bi3+ (1.5 at. %). Did you try to use 2.0 at. % or more?
Author Response
The authors appreciate the time taken by the reviewer to read the manuscript, as well as their comments to improve it.
The whole manuscript needs extensive editing of the English language which makes the readers hard to follow.
I can only mention a few examples
Line 68: thermic treatment
Line 72: synthesized sol-gel method
Line 160-161: I think you mean Figures 4a and 4b
Line192: mayor bismuth
R= All the manuscript has been modified, and an extensive English editing process has been done.
- The synthesis part of the methodology section should be revised.
R= The entire experimental section has been rewritten, in such a way that it can more clearly represent the new proposed sol-gel process, in such a way that it can be reproduced.
- The best intensity was obtained using the higher concentration of Bi3+ (1.5 at. %). Did you try to use 2.0 at. % or more?
R= As now mentioned in the manuscript, it is known that there is only a partial solubility of bismuth in the LuVO4 structure, which is relatively small, as has been shown in other works for YVO4 matrices. On the other hand, XRD results of powders with the same compositions analyzed in the films were added and it was shown that from 1.5% of Bi, the precipitation of a secondary phase begins, apparently of LUVO3. Due to the above, it was not considered convenient to further increase the concentration of bismuth, since it is very likely that the luminescence decayed due to the precipitation of other phases. All the new discussion of the powders as well as the maximum Bi content that can enter the structure has been added from lines 141-158
Reviewer 3 Report
The authors presents a sol-gel synthesis of Eu3+, Bi3+ codoped LuVO4 films.
My major concern with the work is the comparison of the emission intensity that the authors do between different sample. This can only be performed by using a quantitative technique such as measuring the absolute emission quantum yield. The variation that the author measure in Figures 7 and 9 can be due to a change in the thickness of the films. As the author correctly indicated in Figure 5 and 6 the intensity of the spectra is in arbitrary units. And so, the intensity of different spectra should not be done.
Other things that must be improved:
The abbreviation “FTIR” is introduced in line 102 but was already used in lines above.
The sentence that is finished in line 114 need a reference.
In figure 1 “Wavelenght” is misspelled.
In figure 2 “u.a.” needs to be replaced by “a.u.”
The sentence that is finished in line 133 need a reference.
How were determined the crystallite size and texturing coefficients presented in Table 1?
Figure 3 needs a scale.
To prove that the film in Figure 3 is transparent, the author must add a better image. Maybe they can place some text or figure behind the film.
How the size values present in line 165 were determined?
The authors describe the Figure 4 as containing “distributions of circular agglomerations”
This sentence is confused: “Sometimes these particles are interrupted by spaces where no ones are present, however, as shown by the excitation of the films and their emission at the macroscopic level, it is known that the deposit present them on the entire surface. It may be assumed that in areas where no particles exist, smaller or already sintered agglomerations may exist.”
I do not understand how the excitation and emission features of the films at macroscopic level can give information about the distribution of the particles at the microscopic level. What the SEM images indicate is that the particles are in contact between them. A cross section image would provide more information.
The sentence that is finished in line 190 need a reference.
The setup used for the luminescent measurements is not indicated.
Error bars should be added in Figure 7 and 9.
Author Response
The authors appreciate the timely comments of the reviewer, which allowed us to extensively revise the manuscript, add new results to improve it, and in general present better evidence of the transfer mechanism from Bi to Eu. We appreciate the time spent on this evaluation.
The authors presents a sol-gel synthesis of Eu3+, Bi3+ codoped LuVO4 films.
My major concern with the work is the comparison of the emission intensity that the authors do between different sample. This can only be performed by using a quantitative technique such as measuring the absolute emission quantum yield.
R=The reviewer is right, the best way to compare the effects of energy transfer is through the value of the quantum yield. Unfortunately, however, the fluorometer that we have access to does not allow the necessary settings for such experiments. However, to adequately correct the correct observation of the reviewer, new calculations and new experiments were carried out in order to clearly demonstrate that there is an energy transfer between Bi and Eu. First, it has benn calculated the R factor, which determines the degree of asymmetry related to the luminescence of Europium was calculated, and it was found that the emission of the 5D0-7F2 transition, that is, the most anti-symmetric, is favored by the increase in Bi content. of the sample, which confirms that there is indeed such an energy transfer mechanism. Secondly, the energy transfer efficiency was calculated (lines 311 to 324), where the results are explained, in addition to being added in Figure 9b for better understanding. In general, it was also determined that the increase in bismuth increases the transfer efficiency. Also, it has been demonstrated that the exact mechanism of energy transfer occurs via a quadrupole-quadrupole interaction, which is congruent for similar already-published systems. Finally, the life-time decay experiments has been also carried out, which also confirms the energy transfer process, since the life-time of Bi3+, clearly diminishes in presence of Eu 3+ ions. The authors believe that these new evidences clearly demonstrate both the existence of the energy transfer mechanism, as well as the effect that the Bi content has on such process.
The variation that the author measure in Figures 7 and 9 can be due to a change in the thickness of the films.
R=Indeed, the variations in thickness can cause variations in the emission results, however, the authors do not believe that this is the case in this study, because all the deposit conditions were the same for all the samples, with the same solvent compositions in the prepared sols, the only difference being the Bismuth content, so the viscosity, which would be the main parameter that would modify the sol, does not change from composition to composition, so it is acceptable to assume that the thickness is the same. In addition, in order to analyze whether there could be an important variation, 3 Eu-only films were synthesized with the same composition of the sols used, and the emission spectrum was analyzed, where no significant variations were found.
As the author correctly indicated in Figure 5 and 6 the intensity of the spectra is in arbitrary units. And so, the intensity of different spectra should not be done.
R=The reviewer is right in his observation, however, in order to be able to quantify in a better way, and to be able to validate the trends shown in the manuscript, the calculations mentioned above were made, which show that the mentioned trends of increase in luminescence of the eu due to the effect of the Bi are correct.
Other things that must be improved:
The abbreviation “FTIR” is introduced in line 102 but was already used in lines above.
The sentence that is finished in line 114 need a reference.
In figure 1 “Wavelenght” is misspelled.
In figure 2 “u.a.” needs to be replaced by “a.u.”
The sentence that is finished in line 133 need a reference.
R=All changes has been made
How were determined the crystallite size and texturing coefficients presented in Table 1?
R= Both equations has been added to the manuscript (186 y 202 lines) .
Figure 3 needs a scale.
R= The scale has been added
To prove that the film in Figure 3 is transparent, the author must add a better image. Maybe they can place some text or figure behind the film.
R= The corresponding photograph has been added.
How the size values present in line 165 were determined?
R= The particle size was measured directly from the micrographs of 180 individual particles for both samples, in such a way that it was possible to determine the average size. The line has been added to the manuscript
The authors describe the Figure 4 as containing “distributions of circular agglomerations”
This sentence is confused: “Sometimes these particles are interrupted by spaces where no ones are present, however, as shown by the excitation of the films and their emission at the macroscopic level, it is known that the deposit present them on the entire surface. It may be assumed that in areas where no particles exist, smaller or already sintered agglomerations may exist.”
I do not understand how the excitation and emission features of the films at macroscopic level can give information about the distribution of the particles at the microscopic level. What the SEM images indicate is that the particles are in contact between them. A cross section image would provide more information.
R= The entire section was rewritten, because it was not possible to understand what the authors intended, which is that the formation of the agglomeration of the observed particles occurs in certain areas of the film, due to the effect of the energy from heating. In order to better appreciate this assertion, a new SEM photo was added at lower magnifications. On the other hand, the assertion of homogeneity and luminescent tests relationship was totally incorrect on the part of the authors, the idea was simply that since no appreciable differences in the emission of the films were observed macroscopically, it is likely to assume that the coating was found homogeneously. It must be remembered that, when the films are not continuous, the discontinuities are observed in the emission tests. The morphological analysis was, therefore, modified in order to correctly explain the observed phenomena.
The sentence that is finished in line 190 need a reference.
R= The reference has been added
The setup used for the luminescent measurements is not indicated.
R= The set-up has been added in the experimental section.
Error bars should be added in Figure 7 and 9.
R= The bars has been added
Reviewer 4 Report
The authors of the manuscript “Manuscript ID: Materials-1932362, Article Title:
LUMINESCENCE PROPERTIES AND ENERGY TRANSFER 2 OF Eu3+, Bi3+ CODOPED LuVO4 MODIFIED WITH PLURONIC 3 F-127 FILMS OBTAINED BY SOL-GEL “
investigated the luminescence properties of LuVO4 upon co-doping by Bi and Eu ions. The results are interesting and in my view the manuscript can be published in Materials after revision which includes some concerns that should be considered to improve the manuscript.
1- It is difficult to deduce that Bi and Eu doping were occurred in the structure of LuVo4. The EU and Bi can be existed in materials as a result of segregation, or clusters , and etc. The doping (in which Bi or EU is substituted in lattice structure should be more accurately investigated from analyzing the XRD ( the alteration of lattice parameter, or having XPS and analyzing the XPS peaks and its alteration upon doping at different concentration, and etc. I recommend to discuss this part in more details.
2- It would be desired to express that upon doping LuVo4, what do you expect to occur if you are talking about doping. For example please let the reader to know that Bi are expected to substituted at which ( Lu, or V or Oxygen )sites?
3- It was concluded that upon increasing the Bi3+ level, the emission spectra were reduced or in contrast the luminescence spectra are enhanced. How can the authors relate this conclusion to the particle size of each concentration, if there are any relations.

Author Response
The authors appreciate the time taken by the reviewer to read the manuscript, as well as their comments to improve it.
We improved the manuscript as the reviewer suggested:
- It is difficult to deduce that Bi and Eu doping were occurred in the structure of LuVo4. The EU and Bi can be existed in materials as a result of segregation, or clusters , and etc. The doping (in which Bi or EU is substituted in lattice structure should be more accurately investigated from analyzing the XRD ( the alteration of lattice parameter, or having XPS and analyzing the XPS peaks and its alteration upon doping at different concentration, and etc. I recommend to discuss this part in more details.
R= The reviewer is right, because the first version of the manuscript did not clearly explain the Bi substitution phenomenon in the LuVO4 structure. As correctly requested, XRD studies of powders with the same compositions of the films were added, where it is clearly observed that there is clearly a deformation of the crystal lattice due to the incorporation of Bi in LuVO4. However, the presence of other Bi phases is not observed, for example BiVO4 or Bi2O3, which would be the main impurities that could be formed. In the corresponding part of the manuscript, it is also explained that there are references in which it has been determined that Bi can effectively replace Lu, since both have the same valence, and both LuVO4 and BiVO4 are tetragonal, in addition to their difference of atomic radii is not that large. For this reason, for example, in YVO4 it has been found that Bi can be up to 4% before precipitating in the form of BiO4. All the corresponding discussion has been added to the manuscript.
- It would be desired to express that upon doping LuVo4, what do you expect to occur if you are talking about doping. For example please let the reader to know that Bi are expected to substituted at which ( Lu, or V or Oxygen )sites?
R= It is expected that substitutes de Lu positions. The ideas has been added to the manuscript.
- It was concluded that upon increasing the Bi3+ level, the emission spectra were reduced or in contrast the luminescence spectra are enhanced. How can the authors relate this conclusion to the particle size of each concentration, if there are any relations.
R= In the present work, the difference derived from the particle size was not analyzed, since all the films were analyzed in their luminescent properties at 1000 °C, however, we know that, as the particle size decreases, in general, the luminescent intensity tends to increase.
Round 2
Reviewer 1 Report
I thank the authors for the extensive revision of the article based on the report's considerations. I am satisfied with the answers to the questions, and the article has improved in all its sections. There is still some improvement to be made in the English language and style (albeit much improved compared to previous version): I advise the authors to carry out a further check to correct the errors present, for instance:
Page 1, should be "energy transfer" instead of "energy trans";
Page 6, should be "it's due to the movies", not "it's due to the movies";
Page 12, "because of the energy transfer" instead of "because of the tranfer energy".
I still have a question about the lifetime on page 17: Have the lifetime of the 5D0 excited state of the europium (acceptor) been evaluated, to see if they are in line with the data reported in the article of the donor system (bismuth)?
In conclusione, I recommend the publication of the paper in Materials after minor revisions.
Author Response
I thank the authors for the extensive revision of the article based on the report's considerations. I am satisfied with the answers to the questions, and the article has improved in all its sections. There is still some improvement to be made in the English language and style (albeit much improved compared to previous version): I advise the authors to carry out a further check to correct the errors present, for instance:
Page 1, should be "energy transfer" instead of "energy trans";
Page 6, should be "it's due to the movies", not "it's due to the movies";
Page 12, "because of the energy transfer" instead of "because of the tranfer energy".
R= The mentioned corrections were made and the complete file was revised to make some changes
I still have a question about the lifetime on page 17: Have the lifetime of the 5D0 excited state of the europium (acceptor) been evaluated, to see if they are in line with the data reported in the article of the donor system (bismuth)?
R= The decay time of the Eu mono-doped sample was added and, as expected, its decay time is slightly higher than that of the co-doped sample but lower than that of the bi mono-doped sample.
Reviewer 2 Report
Authors successfully answered the all quarries raised. So I recommend this paper for publication.
Author Response
The authors thanks to the revisor about his opinion
Reviewer 3 Report
The authors taken into account my previous comments, but the changes were not satisfactory.
The caption of Figure 3 should indicate what is the technique that allowed the collection of the data.
The authors claim: “On the other hand, it is important to mention that the macroscopic emission studies of the films (Figure 2), do not show zones of preferential emission, from which it is concluded that the entire surface film is covered by LuVO4,” Figure 2 does not give any information about the emission of the films. I believe that the authors are talking about Figure 4 that shows a photography of the samples under UV excitation. I do not think that the data in Figure 4 can be used to discuss if the entire surface of the film is covered. I think that the author do no present data that support statements like this: “Therefore, it could be assumed that in areas where no particles are observed, smaller or already sintered agglomerations exist.” I think that this statement should be removed.
In Figure 6, The 5D0→7F3 is not marked in the spectra
As I claim in my previous report: “The variation that the author measure in Figures 7 and 9 can be due to a change in the thickness of the films.” The authors respond: “the variations in thickness can cause variations in the emission results, however, the authors do not believe that this is the case in this study, because all the deposit conditions were the same for all the samples”. Why do the author not measure the thickness of the films? Besides the thickness, several other variables related with the photoluminescence measurement can affect the intensity of the spectra. Does not make sense to compare the emission intensity of different samples, since the author did not used a quantitative technique.
In line 319 and 320: how the IS and IS0 were determined?
In Figure 10: Can the author add error bars?
In Figure 11: Can the author show the curve that was fitted to the decay curve?
Why authors use equation 6? Can they report the values for the rapid and slow times?
Author Response
The authors again appreciate the reviewer's comments, which are all correct, and allow for better writing, although unfortunately for the authors of the article, some point is not fully resolved due to lack of infrastructure.
The caption of Figure 3 should indicate what is the technique that allowed the collection of the data.
R= In the caption, it has been indicated that the results are from XRD
The authors claim: “On the other hand, it is important to mention that the macroscopic emission studies of the films (Figure 2), do not show zones of preferential emission, from which it is concluded that the entire surface film is covered by LuVO4,” Figure 2 does not give any information about the emission of the films. I believe that the authors are talking about Figure 4 that shows a photography of the samples under UV excitation. I do not think that the data in Figure 4 can be used to discuss if the entire surface of the film is covered. I think that the author do no present data that support statements like this: “Therefore, it could be assumed that in areas where no particles are observed, smaller or already sintered agglomerations exist.” I think that this statement should be removed.
R=As per the reviewer's suggestion, the mentioned analysis has been removed from the article, leaving only clear that similar morphologies have been observed in other vanadate film systems.
In Figure 6, The 5D0→7F3 is not marked in the spectra
R= According to the correct assessment of the reviewer, the aforementioned transition was added.
As I claim in my previous report: “The variation that the author measure in Figures 7 and 9 can be due to a change in the thickness of the films.” The authors respond: “the variations in thickness can cause variations in the emission results, however, the authors do not believe that this is the case in this study, because all the deposit conditions were the same for all the samples”. Why do the author not measure the thickness of the films? Besides the thickness, several other variables related with the photoluminescence measurement can affect the intensity of the spectra. Does not make sense to compare the emission intensity of different samples, since the author did not used a quantitative technique.
R=As the reviewer rightly mentions, there are a lot of variables that can affect the light yield of a film, such as thickness, porosity, phase distribution, densification, etc. Unfortunately, and fully understanding the reviewer's point, now we do not have access to any technique in our institution that allows us to carry out a study of the thickness of the film, such as SEM, since the same one that was used in the present study is found currently out of service. However, as mentioned above, because all the conditions for the synthesis of the sols and their deposition were kept the same, there should not be considerable variations in said properties, which in fact is one of the characteristics of the sol-gel process, in which, as long as the same concentrations of solvents, surface modifiers, pH modifiers, and the same types of metal precursors are maintained, the corresponding deposits will tend to present the same morphological properties. The authors believe that the core part of the work, which is the effect of energy transfer from Bi to Eu, is demonstrated by studying figures 9-11, and although we would like to be able to make a direct measurement of the quantum-yield, our fluorometer does not have the integrating sphere necessary for such a study. Finally, there is a large number of works in the literature where the mentioned effects are reported only with relative intensity values, which effectively, despite not being quantitative, do allow qualitative relationships to be established, for example, the R = I (5D0 →7F2)/I (5D0 →7F1) as a reference to the coordination state and symmetry Furthermore, the energy transfer process is also evident from the decay curves experiments.
In line 319 and 320: how the IS and IS0 were determined?
R= As observed in the aforementioned bibliography, it is possible to obtain said relationship from the emission spectrum of the co-doped samples compared to the monodoped samples.
In Figure 10: Can the author add error bars?
R= Error bars were added, which correspond to 3 independent measurements of the samples. It is worth mentioning that the error is the same, since for the Drexler relation, the values ​​of the ordinates do not change, and what changes is that of the abscissas. Due to the effect of the new calculation, the values ​​of R2 suffered slight modifications in the fourth significant figure, so the analysis carried out does not change.
In Figure 11: Can the author show the curve that was fitted to the decay curve?
The fitted curves has been added, an also, the decay curve of a Eu -modoped sample
Why authors use equation 6? Can they report the values for the rapid and slow times?
R= The adjustment was made by means of the simple curve and the bi-exponential curve, presenting the latter with better adjustments of R2, which also implies that there are two relaxation mechanisms: one due to fluorescence, the fast one, and the other due to structural defects, the slow one. The corresponding text has been added. Also, in the same figure, has been added the rapid and the slow times.
Reviewer 4 Report
The manuscript can be published.
Author Response

(The authors gave the same response as above.)
